# Simplifying Cross-modal Interaction via Modality-Shared Features for RGBT Tracking

## ABSTRACT

Thermal infrared (TIR) data exhibits higher tolerance to extreme environments, making it a valuable complement to RGB data in tracking tasks. RGB-T tracking aims to leverage information from both RGB and TIR images for stable and robust tracking. However, existing RGB-T tracking methods often face challenges due to significant modality differences and selective emphasis on interactive information, leading to inefficiencies in the cross-modal interaction process. To address these issues, we propose a novel Integrating Interaction into Modality-shared Fearues with ViT(IIMF) framework, which is a simplified cross-modal interaction network including modality-shared, RGB modality-specific, and TIR modality-specific branches. Modality-shared branch aggregates modality-shared information and implements inter-modal interaction with the Vision Transformer(ViT). Specifically, our approach first extracts modality-shared features from RGB and TIR features using a cross-attention mechanism. Furthermore, we design a Cross-Attention-based Modality-shared Information Aggregation (CAMIA) module to further aggregate modality-shared information with modality-shared tokens. We evaluate our model on three widely-used benchmark datasets and extensive experiments demonstrate that our method achieves state-of-the-art performance. All the source code and trained models will be released.

## CCS CONCEPTS

• **Computing methodologies → Tracking**.

## KEYWORDS

RGB-T tracking, three-branch Vision Transformer, inter-modal interaction

## 1 INTRODUCTION

Single Object Tracking (SOT) aims to localize a target object in a video sequence given its initial position in the first frame, which has garnered significant attention due to its wide range of applications, such as intelligent robotics, autonomous driving, and video surveillance. Despite considerable progress in recent years, many SOT methods [7, 42] rely on visible images and suffer from performance degradation under challenging conditions, such as stormy weather and low-illumination environments. To address these limitations, RGB-T tracking has emerged as a promising solution, leveraging

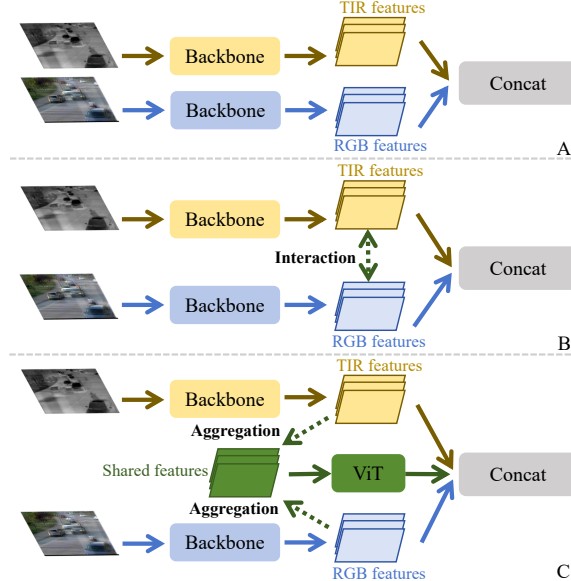

**Figure 1: Comparison between current RGB-T tracking methods and our method. In part A, features of RGB and TIR images are concatenated directly. In part B, models bridge the interaction between features and concatenate them at last. In part C, our approach extracts modality-shared features, and bridges the interaction of two modality information with ViT.**

the complementary information present in both visible and thermal infrared (TIR) images to enhance tracking performance across various scenarios.

The key challenge in RGB-T tracking lies in the effective interaction and exploitation of complementary information between RGB and TIR modalities. Existing approaches can be broadly categorized into two groups: direct feature concatenation and feature interaction. As illustrated in Fig 1 A, some methods [31, 32, 45] directly concatenate features from both modalities before feeding them into the prediction network. However, this naive concatenation strategy underestimates the importance of inter-modality interaction and may inadvertently incorporate substantial background noise, compromising the effectiveness of feature fusion and impairing tracking performance. In contrast, as shown in Fig 1 B, other RGB-T methods [15] attempt to bridge interactions between RGB and TIR features using various techniques, such as employing templates as mediums to distribute information between modalities. Nevertheless, using a fused template may lead to an overemphasis on modality-shared features at the expense of modality-specific information, while using separate templates for each modality may

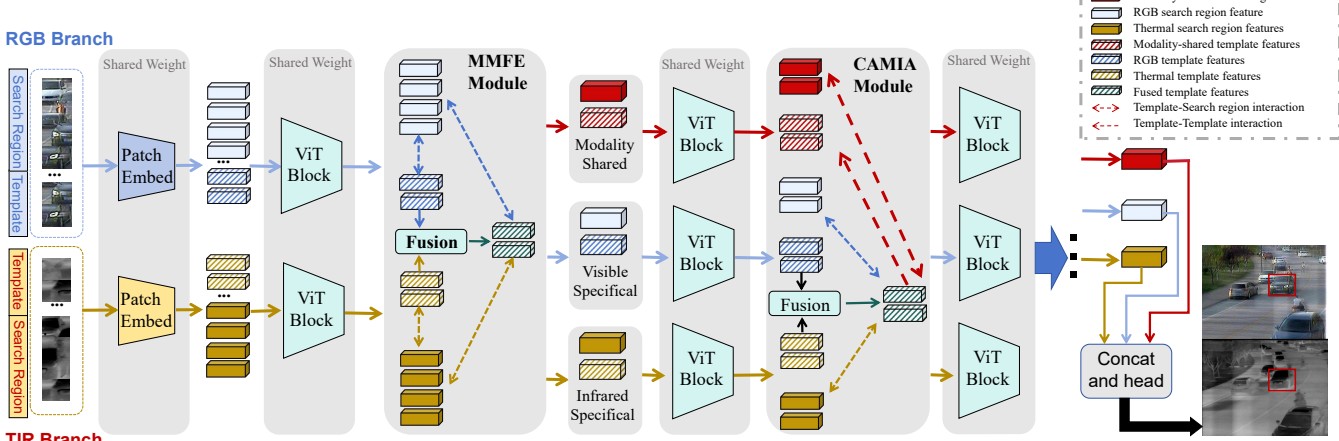

**Figure 2: Overall framework of our proposed IIMF algorithm. The IIMF algorithm processes RGB and TIR images using ViT blocks to extract features. These features are then input into the MMFE module, where both modality-shared and modality-specific features are extracted through interactions between RGB and TIR tokens, facilitated by template tokens. Subsequently, the CAMIA module aggregates target-relevant, modality-shared features and assigns these to modality-shared features. Finally, all search region features are consolidated and input into the prediction head for target state prediction.**

result in inefficient communication due to significant differences in their characteristics.

To tackle these challenges, we propose a novel simplified modality fusion method for RBG-T tracking by Integrating Interaction into Modality-shared Feature(IIMF) which employs modality-shared features to collect the modality-shared information from two modalities and bridges interaction between two modality information within ViT block as shown in Fig 1 C. Unlike existing methods, IIMF facilitates interaction between RBG and TIR modalities by directly feeding the modality-specific and modality-shared features into ViT block without complicated fusion methods. This architecture allows each modality-specific branch to focus on extracting and preserving valid information unique to its respective modality, while the modality-shared branch effectively aggregates information common from both modalities. By introducing a dedicated modality-shared feature, IIMF ensures the effective collection and representation of modality-shared information while preserving valuable modality-specific details that might be overlooked in a naive fusion scheme. This strategic separation and targeted aggregation of features enhance the model's representational capacity, leading to improved tracking performance in challenging multi-modal scenarios.

Specifically, we design a Modality-shared and Modality-specific Feature Extraction (MMFE) module to extract the modality-shared ($X_{sh}$) and modality-specific ($X_v$ for RGB and $X_i$ for TIR) features. The MMFE module employs the fused template as a query and the RGB and TIR search regions as keys and values to aggregate modality-shared information and obtain the multi-modal context medium via a cross-attention operation. This medium is then used as a key and value, with the RGB and TIR search regions as queries, to distribute modality-shared information to both search regions and fuse them, resulting in $X_{sh}$ with enhanced target-relevant information. Additionally, cross-attention is also applied to $X_v$ and

$X_i$, using their respective templates as mediums to collect modality-specific target-relevant information from their corresponding search regions. $X_v$ and $X_i$ serve as modality-specific features in the subsequent network.

To further aggregate modality-shared information to $X_{sh}$, we also design the Cross-Attention-based Modality-shared Information Aggregation(CAMIA) module. CAMIA first obtains a multi-modal context medium by employing the fused template of RGB and TIR templates as a query in a cross-attention mechanism with $X_v$ and $X_i$ as keys and values. Furthermore, this medium serves as a key and value in a cross-attention mechanism with $X_{sh}$ and $Z_{sh}$ as the query, enabling the distribution of modality-shared information to $X_{sh}$ and $Z_{sh}$. By implementing the CAMIA module, more modality-shared information is aggregated to $X_{sh}$, which benefits the information interaction between RGB and TIR search region within the ViT block.

To evaluate the performance of our method, we conduct experiments on three commonly used RGB-T benchmark datasets, including RGBT210 [23], RGBT234 [20], and LasHeR [22]. Experimental results show that our model achieves state-of-the-art performance, showcasing its ability to achieve stable and robust tracking performance. The main contributions of this work are summarized as follows:

- We propose a novel and simplified RGB-T Tracking method, enabling the implementation of inter-modal interaction in ViT block, which replaces the complicated inter-modal interaction.
- We propose a novel MMFE module to disentangle features into modality-shared and modality-specific components, enabling the aggregation of modality-shared information while preserving modality-specific details.
- Our method achieves state-of-the-art performance on several RGB-T tracking benchmarks. We also conduct extensive

experiments including ablation studies to demonstrate the effectiveness of the proposed method and the effect of every component.

## 2 RELATED WORKS

### 2.1 Single Object Tracking

Single object tracking(SOT) works as one of the fundamental tasks in the field of computer vision. It aims to continuously localize the target object within the sequences and serves as the downstream work of many other computer vision tasks. Great progress has been made in the field of SOT for accurate and stable target object tracking in various scenarios. Siamese-based methods [1, 8, 18, 19, 38, 50, 51, 58] focus on computing the correlation between template and search region to realize the tracking process. Bertinetto et al. [1] firstly introduces the Siamese network to the field of visual object tracking. It used AlexNet to work as the backbone of the model and used cross-correlation operations to get the response map. Some online methods [2, 7, 13, 16, 31] aim to enhance the accuracy and robustness of the tracking algorithms with an on-line updating mechanism that dynamically adjusts to the evolving characteristics of the target. With the success of transformer in the field of computer vision, Some works [4, 6, 17, 37, 41–43] devoted to introducing ViT to tracking task, which employs vision transformer(ViT) to work as the backbone of their models and use self-attention and cross-attention for interaction between search region and template features which integrate the information of search region and template for matching relationship modeling. Ye et al. [42] integrate template and search region features and feed it into a one-stream backbone which is ViT for jointly feature learning and relationship modeling and their OSTracker gains great sucess.

### 2.2 RGB-T Tracking

General SOT methods always only focus on visible object tracking(VOT), so they would encounter catastrophic performance degradation if the sequences are captured under extreme conditions. Therefore, some works [11, 15, 25, 27, 39, 47, 52, 52, 53, 56] attempt to jointly utilize visible and thermal infrared images for high-performance tracking since thermal infrared data could furnish supplementary information under some extreme conditions. Zhang et al. [47] present a novel framework aiming at improving the efficiency and accuracy of RGB-T tracking algorithms. They introduce a cross-modality distillation framework to bridge the performance gap between compact and powerful trackers. The proposed Specific-Common Feature Distillation (SCFD) module transforms both modality-common and modality-specific information from a deeper two-stream network to a shallower one-stream network. Zhang et al. [45] propose a novel framework to fuse RGB and TIR features in the context of tracking and embed it into the DiMP [3] tracker for RGB-T tracking. Hui et al. [15] integrate a Template-Bridged Search Region Interaction(TBSI) module into ViT backbone to exploit templates as a bridge for cross-modal interaction. They identify that the previous methods might introduce redundant background noise or limit the RGB and TIR modal interaction to local regions. The TBSI module allows for high performance of search region interaction and the original templates to

be updated with enriched multi-modal contexts which further improve the performance of tracking. Our method follows their idea of employing templates to bridge the cross-modal interaction and insert our own novel idea for better performance. Apart from the framework research, the research of the RGB-T dataset witnesses substantial development [14, 20, 22, 23]. Li et al. [22] propose the LasHeR dataset which is currently the largest RGB-T Tracking dataset.

## 3 METHOD

### 3.1 Overview

The overall framework of our method is illustrated in Fig 2. Specifically, to simplify the cross-modal interaction, We employ a Vision Transformer(ViT) as backbone of our model, which also assuming the responsibility of removing the carefully designed cross-modal interaction modules. Following the operations of ViT, the input RGB and TIR images are first split into patches and fed into the ViT backbone. Afterward, our proposed Modality-shared and Modality-specific Feature Extraction(MMFE) module is seamlessly integrated into the ViT backbone to aggregate modality-shared target-relevant information and bridge the interaction between the visible and thermal infrared search regions. Furthermore, the Cross-Attention-based Modality-shared Information Aggregation(CAMIA) module further aggregates modality-shared information from RGB and TIR search regions after partial feature learning into modality-shared tokens. Finally, the search region features from modality-shared and modality-specific branches are concatenated and fed into the tracking head for target object state prediction, enabling robust and accurate tracking in challenging multi-modal scenarios.

### 3.2 Simplified Cross-modal Interaction

Inspired by the impressive performance of Vision Transformers (ViTs) [9] in object tracking [42], we simplify the cross-modal interaction by directly integrating modality-shared and modality-specific features into ViT without complicated modality fusion methods. This new structure is aimed at the simultaneous extraction of features that are shared across different modalities as well as those specific to each modality, following recent high-performance tracking methods [15].

Let $I_v^x \in \mathbb{R}^{H_x \times W_x \times 3}$, $I_i^x \in \mathbb{R}^{H_x \times W_x \times 1}$ denote RGB and TIR search region images, and $I_v^z \in \mathbb{R}^{H_z \times W_z \times 3}$, $I_i^z \in \mathbb{R}^{H_z \times W_z \times 1}$ denote RGB and TIR template images respectively. Following the operation of ViT, we initially partition these images into patches of size $P \times P$. These patches are then transformed into sequences of embedded features, hereafter referred to as *tokens*. Specifically, for the RGB and TIR modalities of the search region, their tokens are denoted as $X_v$ and $X_i$, respectively, with dimensions $\mathbb{R}^{N_x \times C}$. For the RGB and TIR modalities of the template images, the tokens are denoted as $Z_v$ and $Z_i$, respectively, with dimensions $\mathbb{R}^{N_z \times C}$. Here, $N_x = H_x \times W_x / P^2$ and $N_z = H_z \times W_z / P^2$ denote the patch number of search region and template, and $C$ represents the number of channels in each token. This structured approach allows for efficient embedded feature extraction, which can be represented as

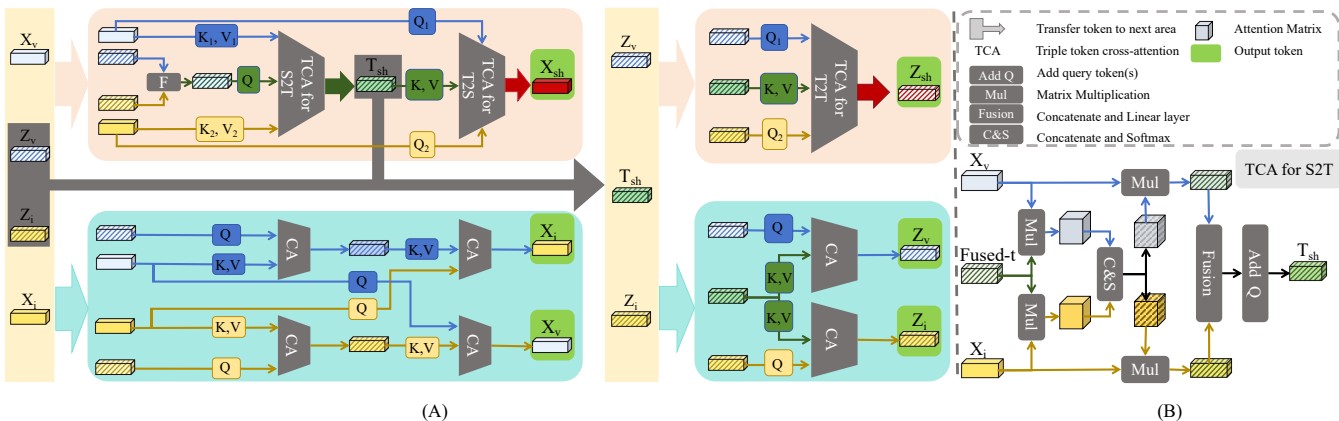

(A)

(B)

**Figure 3: Illustration of our MMFE module. The network whose background is red is used for modality-shared tokens generation. The network whose background is green is used for updating modality-specific tokens. CA for S2T, T2S, and T2T are similarly conducted.**

the following formula:

$$X_v = L(P(I_v^x)), \quad Z_v = L(P(I_v^z))$$
$$X_i = L(P(I_i^x)), \quad Z_i = L(P(I_i^z)) \tag{1}$$

where $L$ denotes the linear layer function and $P$ denotes the function of splitting and flattening. RGB and TIR search region and template images share the same patch embedding operation.

Afterward, we concatenate search region and template tokens as joint embedded tokens $H_r = [Z_r; \; X_r] \in \mathbb{R}^{(N_x+N_z)\times C}$. These tokens are then processed through the ViT to learn features and model relationships jointly. The operations for the Thermal Infrared (TIR) branch are the same as the RGB branch. The two-modality network is transformed into a three-stream network after the processing of the Modality-shared and Modality-specific Feature Extraction (MMFE) module. The formula is shown as follows:

$$X_v, \; X_i, \; X_{sh} = MMFE([X_v; \; X_i]) \tag{2}$$

where $MMFE$ denotes the function carried out by the MMFE module, with each branch extracting features specific to its modality—RGB, TIR, and shared modalities, respectively (details described in Section 3.3).

Additionally, the modality-shared branch plays a crucial role in enabling cross-modal information interaction. To clarify the operation within the ViT blocks, we describe it using the following formula:

$$\begin{aligned}O &= LN(\sigma(QK^T)V)\\ &= LN(\sigma([Q_z; \; Q_x][K_z; \; K_x]^T)V)\\ &= LN(\sigma([Q_zK_z^T, \; Q_zK_x^T; \; Q_zK_z^T, \; Q_xK_x^T])[V_z;V_x])\\ &= LN(\sigma([Q_zK_z^TV_z + Q_zK_x^TV_x; \; Q_xK_z^TV_z + Q_xK_x^TV_x]))\end{aligned} \tag{3}$$

where $LN$ denotes the layer normalization function, $\sigma$ is the softmax function, and $O$ is the output.

Further, the cross-attention mechanism is generalized by constructing adaptive weighted summation operations on $X_v$ and $X_i$,

allowing us to express the modality-shared tokens $X_{sh}$ and $Z_{sh}$ as:

$$X_{sh} = W_v^x X_v + W_i^x X_i$$
$$Z_{sh} = W_v^z Z_v + W_i^z Z_i \tag{4}$$

where $W$ denotes the weight matrix, the superscripts $x$ and $z$ denote search region and template respectively, and the subscripts $v$ and $i$ denote visible and infrared modality respectively. Taking the component $Q_z K_x^T V_x$ from Formula 3, we can get the formulas as follows by incorporating Formula 4.

$$\begin{aligned}Q_z K_x^T V_x =&(WZ_v + WZ_i)(WX_v + WX_i)^T(WX_v + WX_i)\\ =&WZ_v(WX_v)^T WX_v + WZ_i(WX_i)^T WX_i+\\ &WZ_i(WX_v)^T WX_v + WZ_v(WX_i)^T WX_i+\\ &WZ_v(WX_v)^T WX_i + WZ_i(WX_i)^T WX_v+\\ &WZ_i(WX_v)^T WX_i + WZ_v(WX_i)^T WX_v\end{aligned} \tag{5}$$

and another component $Q_x K_x^T V_x$ is shown as follows:

$$\begin{aligned}Q_x K_x^T V_x =&(WX_v + WX_i)(WX_v + WX_i)^T(WX_v + WX_i)\\ =&WX_v(WX_v)^T WX_v + WX_i(WX_i)^T WX_i+\\ &WX_v(WX_i)^T WX_i + WX_i(WX_v)^T WX_v+\\ &WX_v(WX_i)^T WX_v + WX_i(WX_v)^T WX_i+\\ &WX_i(WX_i)^T WX_v + WX_v(WX_v)^T WX_i\end{aligned} \tag{6}$$

where subscripts and superscripts of $W$ are omitted for brevity. The Formula 5 and 6 aim to extract the inter- and cross-modal features of the search regions, where $WX_v(WX_v)^T WX_v + WX_i(WX_i)^T WX_i$ is the self-attention operation for RGB and TIR search regions, and the component $WX_v(WX_i)^T WX_i + WX_i(WX_v)^T WX_v$ serves as the cross-attention between RGB and TIR search regions. Obviously, the component $WZ_v(WX_v)^T WX_v + WZ_i(WX_i)^T WX_i$ is an enhancement for target-relevant information. The weight matrix $W$ could also serve as the projection layers of the self-attention and cross-attention mechanism. Therefore, it indicates that aggregating

the information of both RGB and TIR information into modality-shared tokens and feeding them into ViT blocks implements not only intra-modal attention for $X_v$ and $X_i$ but also cross-modal attention between $X_v$ and $X_i$. Considering that the cross-attention mechanism is widely used for inter-modal interaction, our operation of feeding modality-shared tokens into ViT has the ability of inter-modal information interaction. Consequently, it is validated that substituting the direct inter-modal interaction between RGB and TIR modalities with aggregating information into modality-shared tokens and feeding them into the ViT block is significant.

By employing this simple cross-modal interaction architecture, our model adaptively extracts both modality-specific and modality-shared features, enhancing the overall capacity for inter-modal interaction and retaining more target-relevant information across different sensing modalities.

## 3.3 Modality-shared and Modality-specific Feature Extraction

To better extract modality-shared features, we propose our Modality-shared and Modality-specific Feature Extraction(MMFE) module incorporating a cross-attention mechanism to enhance feature extraction for robust tracking. As illustrated in Fig 3, the MMFE module first employs a fused template as the medium to aggregate modality-shared information, denoted as $T_{sh}$. $T_{sh}$ contains rich target-relevant information which is subsequently distributed to both the visible (RGB) and thermal infrared (TIR) search region tokens, $X_v$ and $X_i$, respectively. Afterward, we extract the fused features between $X_v$ and $X_i$. At last, we generate modality-shared template token $Z_{sh}$ and update the template $Z_v$ and $Z_i$ by fusing the features among $Z_v$, $Z_i$, and $T_{sh}$.

Specifically, MMEF initially extracts the fused template features, denoted as $X_{sh}$, among $X_v$ and $X_i$ to aggregate modality-shared information. We adjust the cross-attention module to enable simultaneous processing of three tokens, which is named TCA in Fig 3. We first introduce TCA for S2T to aggregate modality-shared target-relevant information onto the fused template. Since $X_v$ and $X_i$ both serve as key and value in cross attention with a fused template as query, the attention matrices computed by $fused_t \times X_v^T$ and $fused_t \times X_i^T$ are both used to measure the similarity with the same query token, where $fused_t$ denotes the fused template features. Therefore, we concatenate the two attention matrices and perform the function of softmax on them together as shown in Fig 3 (B). The formula is shown as follows:

$$A_v = fused_t \times X_v^T$$
$$A_i = fused_t \times X_i^T \qquad (7)$$
$$[\widetilde{A_v}; \widetilde{A_i}] = \sigma([A_v; A_i])$$

where $A_v$, $A_i$ denote the attention matrix of $X_v$ and $fused_t$, $X_i$ and $fused_t$ respectively, $\times$ denotes matrix multiplication, $\widetilde{A}$ denotes the attention matrix after softmax. The joint softmax operation facilitates the capacity of the model to obtain a richer context since it takes information both from RGB and TIR search regions into a comprehensive account. Afterward, $X_v$ and $X_i$ multiply their own attention matrix separately and we fuse the results to get the output token. Drawing from the concept of residual learning, we ultimately add the query token to the output token and get $T_{sh}$ which is rich

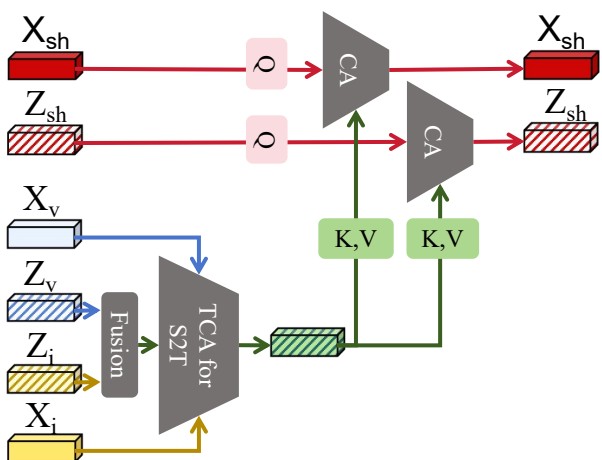

Figure 4: Illustration of CAMIA module. We omit the detail of cross-attention operation since it is a widely-used mechanism. TCA for T2T is similar to TCA for S2T in the MMFE module.

in modality-shared information of both RGB and TIR modalities. The formula is shown as follows:

$$T_{sh} = Query + L(Conc([T_v; T_i]))$$
$$= Query + L(Conc([A_v \times Value_v; A_i \times Value_i])) \qquad (8)$$

where $Query$, $Value_v$, $Value_i$ denote $fused_t$ which serves as query, visible search region token which serves as value, thermal infrared search region token which serves as value, $L$, $Conc$ denote the function of linear layer and concatenating respectively. We then construct TCA on $T_{sh}$, $X_v$ and $X_i$ to generate $X_{sh}$. The operation of TCA for T2S, as is shown in Fig 3 (A), is similar to TCA for S2T, and further elaboration is omitted for brevity. Compared to the approach of performing cross-attention separately before fusion, our method embeds the fusion process into a cross-attention process, which integrates the information of the search region tokens from both modalities, resulting in a more refined fusion scheme when combining RGB and TIR search region tokens.

In addition to the extraction of modality-shared features, we also enhance the target-relevant information for modality-specific tokens. Although RGB and TIR modality-specific branches are employed to preserve modality-specific information, we also perform inter-modal interaction on them to reduce inter-modal differences. As is shown in Fig 3 (A), To update $X_i$, we employ $Z_v$ to aggregate modality-specific information of $X_v$ and distribute it to $X_i$ with cross attention mechanism. The operation for updating $X_v$ is the same.

At last, we generate $Z_{sh}$ by performing TCA for T2T on $fused_t$, $Z_v$ and $Z_i$, where $Z_{sh}$ denotes modality-shared template token which is used for joint feature extraction for modality-shared search region token. And we update $Z_v$ by conducting cross attention on $T_{sh}$ and $Z_v$, which is similar to $Z_i$.

## 3.4 Cross-Attention-based Modality-shared Information Aggregation

For further modality-shared information aggregation, we propose our Cross-Attention-based Modality-shared Information Aggregation(CAMIA) module. The architecture of our CAMIA is shown in Fig 4. We integrate our CAMIA module into our ViT backbone after the 4th block.

Firstly, to aggregate modality-shared information, we fuse $Z_v$ and $Z_i$ as a medium to collect information from both $X_v$ and $X_i$ with TCA for S2T. Then, we perform cross attention between the intermediate template, rich in search region modality-shared information, and $X_{sh}$ to distribute the modality-shared information to $X_{sh}$. We also update the modality-shared template token with the intermediate template by performing cross-attention.

To preserve the modality-specific information, inter-modal interactions are not implemented in the subsequent stages of our network. Although direct interactions are not adopted, the information of RGB and TIR search regions is aggregated into modality-shared search region token $X_{sh}$. The interaction between RGB and TIR information is performed by conducting self-attention on $X_{sh}$. The overall operation of the CAMIA module can be represented by the formula shown as follows:

$$
\begin{aligned}
T_{fused} &= L([Z_i;\ Z_i]) \\
T_{inte} &= TCA([T_{fused};\ X_v;\ X_i]) \\
\widetilde{X_{sh}} &= CA([T_{inte};\ X_{sh}]) \\
\widetilde{Z_{sh}} &= CA([T_{inte};\ Z_{sh}])
\end{aligned}
\tag{9}
$$

where $T_{fused}$ denotes the fused template, $T_{inte}$ denotes the intermediate template, $L$, $TCA$ and $CA$ denote the function of linear layer, triple token cross attention and cross attention, $\widetilde{X_{sh}}$ and $\widetilde{Z_{sh}}$ denote updated $X_{sh}$ and $Z_{sh}$. At last, we realize the aggregation of modality-shared information. The interaction between RGB and TIR modality-specific branches is implemented by aggregating information into the modality-shared token and feeding it into ViT. The modality-specific information is well preserved since no direct interaction is performed.

## 3.5 State Estimation and Training Objective

**State Estimation.** To predict the current state of the target object in the search regions, we adopt the common practice of tracking head possessing classification, bounding box center, and bounding box size prediction branches. The three branches share the same architecture comprising 4 Conv-BN-ReLU layers. Each patch constitutes an anchor since we partition the images into patches for ViT. The classification branch outputs classification score maps Cls for selecting anchor, and the bounding box center and size prediction branches output offset maps O for compensating the reduction in resolution and size maps S for measuring the size of the bounding box. The computation of the state is as shown in the following formula:

$$
[x,\ y,\ w,\ h] = [x_{cls} + O_x,\ y_{cls} + O_y,\ S_w,\ S_h]
\tag{10}
$$

where $[x,\ y,\ w,\ h]$ denotes the bounding box format of our model, $(x_{cls},\ y_{cls})$ denotes the center coordinates of the anchor,

$(O_x,\ O_y)$ denotes the offset from anchor center to bounding box center, $(S_w,\ S_h)$ denotes the bounding box size.

**Training Objective.** Following the training objective of TBSI [15], our loss is computed by focal loss [24] for classification, $L1$ loss [12] for offset, and GIoU loss [34] for bounding box size. The overall loss function is shown as follows:

$$
Loss = L_{focal} + \lambda_{GIoU}L_{GIoU} + \lambda_{L1}L_{L1}
\tag{11}
$$

where $\lambda_{GIoU}$ equals 2.0 and $\lambda_{L1}$ equals 5.0 in our training process.

## 4 EXPERIMENTS

### 4.1 Datasets and Evaluation Metrics

We conduct experiments on three prevailing RGB-T tracking benchmarks, including RGBT210 [23], RGBT234 [20], and LasHeR [22]. The RGBT210 dataset, first introduced in [23], comprises 210 video pairs, totaling approximately 210,000 frames, with the longest video encompassing about 8,000 frames, making it suitable for long-term tracking studies. The RGBT234 dataset, an extension of RGBT210, contains 234 video pairs with around 234,000 frames and is detailed in [20]. Meanwhile, the LasHeR dataset, described in [22], includes 1,224 annotated video sequences, 245 of which are designated for testing. To assess our model comprehensively, we utilize three widely recognized metrics: Success Rate (SR), Precision Rate (PR), and Normalized Precision Rate (NPR). PR is defined as the percentage of frames where the predicted bounding box is within a specified threshold from the ground truth. NPR normalizes the precision rate on the size of the ground truth bounding box to reduce sensitivity to the size of the target object. SR denotes the percentage of the frames whose overlap ratio between the bounding box of output and ground truth is larger than a threshold.

### 4.2 Implementation Details

Our model is implemented utilizing PyTorch and trained on 4 NVIDIA V100 GPUs. Search region images are reshaped to 256×256 and template images are reshaped to $128 \times 128$. The patch size of images is $16 \times 16$. We train our model on the LasHeR training dataset and test on the LasHeR testing dataset, RGBT210 dataset, and RGBT234 dataset without further fine-tuning. The ViT backbone is pre-trained on the SOT dataset. The learning rate is set to $1 \times e^{-4}$ and decayed to 10% after 10 epochs. The training epoch is set to 25. Our MMFE module is inserted into ViT after the 2nd ViT block and the CAMIA module is inserted after the 4th ViT block. The threshold for metrics is set to 20 pixels as a common practice.

### 4.3 Comparison with State-of-the-art Methods

We compare our method with previous state-of-the-art methods on the LasHeR, RGBT210, and RGBT234 datasets. As shown in Table 1, which presents the result of our method testing on the LasHeR dataset, our method outperforms previous methods on all metrics, demonstrating the effectiveness of our method. Compared with the TBSI method, which also employs the ViT-base as the backbone and integrates their module into ViT, our method gains 3.2%/2.7%/2.5% improvement in PR/NPR/SR respectively. For a fair comparison, we also adopt SOT pretraining which is the same as the TBSI method. Furthermore, our IIMF-MMFE method, which

**Table 1: The comparison between our method and previous state-of-the-art methods on the LasHeR dataset. The best and the second results are colored red and blue respectively.**

| Method | Pretraining | Framework | LasHeR | | |
| --- | --- | --- | --- | --- | --- |
| | | | Precision | Norm Precision | Success |
| FANet [54] | ImageNet | CNN | 44.1 | 38.4 | 30.9 |
| CAT [21] | ImageNet | CNN | 45.0 | 39.5 | 31.4 |
| MaCNet [44] | - | CNN | 48.2 | 42.0 | 35.0 |
| DMCNet [30] | ImageNet | CNN | 49.0 | 43.1 | 35.5 |
| APFNet [40] | ImageNet | CNN | 50.0 | 43.9 | 36.2 |
| FTNet [5] | - | CNN | 52.6 | - | 38.1 |
| mfDiMP [45] | SOT | CNN | 59.9 | - | 46.7 |
| QAT [26] | - | CNN | 64.2 | 59.6 | 50.1 |
| STMT [35] | - | ViT | 67.4 | 63.4 | 53.7 |
| TBSI [15] | SOT | ViT | 69.2 | 65.7 | 55.6 |
| IIMF-MMFE | SOT | ViT | 71.3 | 67.6 | 57.3 |
| IIMF | SOT | ViT | 72.4 | 68.4 | 58.1 |

**Table 2: The comparison between out method and previous state-of-the-art methods on RGBT234 dataset. The best results are indicated in bold.**

| Method | RGBT234 | |
| --- | --- | --- |
| | Precision | Success |
| MDNet+RGBT [31] | 72.2 | 49.5 |
| SiamCDA [48] | 76.0 | 56.9 |
| MaCNet [44] | 76.4 | 53.2 |
| MANet [28] | 77.7 | 53.9 |
| FANet [55] | 78.7 | 55.3 |
| MANet++ [29] | 79.5 | 55.9 |
| CAT [21] | 80.4 | 56.1 |
| ADRNet [46] | 80.7 | 57.0 |
| SiamIVFN [33] | 81.1 | 63.2 |
| CMPP [36] | 82.3 | 57.5 |
| APFNet [40] | 82.7 | 57.9 |
| DMCNet [30] | 83.9 | 59.3 |
| mfDiMP [45] | 84.2 | 59.1 |
| SiamIVFN [33] | 81.1 | 63.2 |
| TBSI [15] | 87.1 | 63.7 |
| SiamAfb [10] | **89.0** | 60.2 |
| IIMF | 86.8 | **64.4** |

**Table 3: The comparison between other methods and previous methods on the RGBT210 dataset. The best results are indicated in bold.**

| Method | RGBT234 | |
| --- | --- | --- |
| | Precision | Success |
| DSiamMFT [49] | 64.2 | 62.5 |
| TFNet [57] | 77.7 | 52.9 |
| CAT [21] | 79.2 | 53.3 |
| DMCNet [30] | 79.7 | 55.5 |
| mfDiMP [45] | 84.9 | 59.3 |
| TBSI [15] | 85.3 | **62.5** |
| IIMF | **85.6** | 62.4 |

integrates only MMFE modules into three-branch ViT, also exhibits superior performance, gaining 2.1%/1.9%/1.7% improvement in PR/NPR/SR respectively compared to TBSI, which demonstrates that integrating modality-shared information into modality-shared token and implementing inter-modal interaction with the token fed into ViT can replace the direct inter-modal interaction between both modality features. Compared to the CNN-based methods, our method gains a great improvement in all metrics, which demonstrates that the feature learning and relationship modeling capabilities of ViT make sense in the field of SOT. The comparison with the previous state-of-the-art methods demonstrates that our model achieves valuable and indispensable improvement in both tracking

performance and efficiency. Table 2 and Table 3 present the testing results on RGBT210 and RGBT234 respectively, illustrating that our model maintains good performance on small datasets.

## 4.4 Ablation Studies

To validate the effectiveness of our modules, we conduct ablation studies on the LasHeR dataset with SOT pertaining. The results are shown in Table 4.

**Baseline** method employs the ViT as the backbone and extends it to a three-branch network. The three branches share architecture and weights, utilized separately for the modality-shared, RGB modality-specific, and TIR modality-specific features. The modality-shared feature is generalized by concatenating RGB and TIR features. No interaction among the three branches is implemented. The output features of the three branches are concatenated and fed into the tracking head. By comparing the performance of previous state-of-the-art methods and the Three-branch Baseline, we can conclude that aggregating the information of RGB and TIR search regions into modality-shared tokens benefits feature learning. The improvement can serve as empirical evidence supporting the assertion that the information of RGB and TIR could well interact by

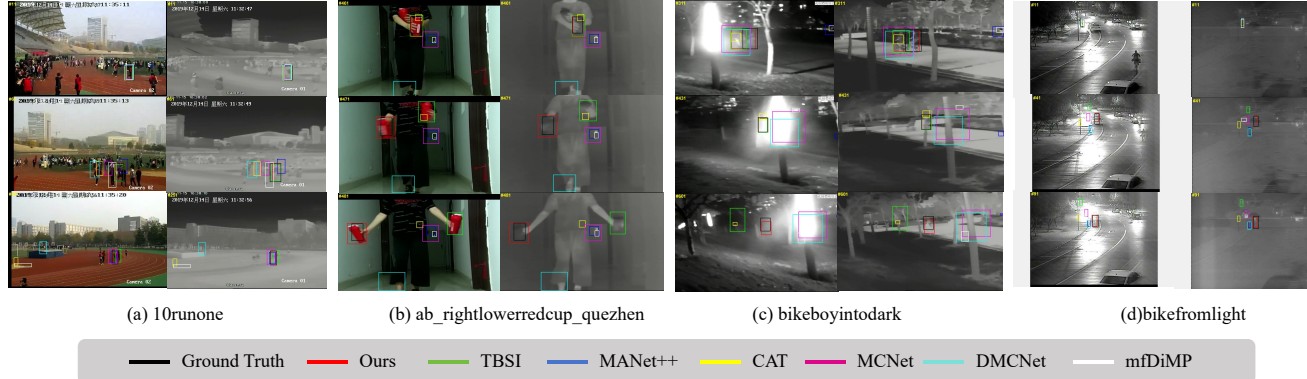

| (a) 10runone | (b) ab_rightlowerredcup_quezhen | (c) bikeboyintodark | (d)bikefromlight |

Ground Truth — Ours — TBSI — MANet++ — CAT — MCNet — DMCNet — mfDiMP

Figure 5: Qualitative comparison between our tracker and other RGB-T trackers on the LasHeR dataset.

Table 4: The ablation studies on the components of our proposed IIMF model. The best results are indicated in bold.

| Method | PR | NPR | SR |
|---|---|---|---|
| IIMF-Baseline | 70.1 | 66.3 | 56.2 |
| IIMF-MMFE | 71.3 | 67.6 | 57.3 |
| IIMF-Full | **72.4** | **68.4** | **58.1** |

feeding the modality-shared tokens into the ViT backbone as we state in Section 3.2.

**IIMF-MMFE** denotes the model only integrating the MMFE module into the three-branch baseline. The MMFE module is inserted after the second ViT Block to extract modality-shared tokens and modality-specific tokens. The backbone before MMFE is a two-branch network while the backbone after MMFE is transformed into a three-branch network. As shown in Table 4, IIMF-MMFE gains 1.2%/1.3%/1.1% improvement in PR/NPR/SR, demonstrating that modality-shared and modality-specific features are well extracted. The modality-shared information is aggregated and the interaction between the information of the two modalities is implemented by feeding the modality-shared token into the ViT backbone. At the same time, more modality-specific information is preserved.

**IIMF-Full** denotes the model integrating the MMFE and CAMIA modules into the three-branch baseline. The MMFE module is inserted after the second ViT block to extract modality-shared tokens and modality-specific tokens, and the CAMIA module is inserted after the 4th ViT block to further aggregate modality-shared information to modality-shared tokens. IIMF-Full gains 1.1%/0.8%/0.8% improvement in PR/NPR/SR compared to IIMF-MMFE, demonstrating that our CAMIA module can further aggregate modality-shared information, which becomes manifest after partial feature learning.

### 4.5 Analysis and Visualization

To demonstrate the effectiveness of our method, we conduct visualization work.

**Qualitative Comparison.** As shown in Fig 5, we conduct a qualitative comparison between our IIMF tracker and 8 other trackers on some challenging scenarios, including fast movement, complex

background, low illumination, high illumination, *e.g.* from LasHeR dataset. For example, the target bike going into the dark in the (c) sequence experiences intense changes in illumination, while our tricker still localizes the target bike since we can leverage the information of RGB and TIR images. The data from TIR modality is a valuable complementary to RGB data under extreme illumination and these results indicate that our method exhibits outstanding performance in the realm of cross-modal data interaction. Furthermore, in the (a) sequence, the target person is often occluded by others, while our tracker localizes the target successfully, which demonstrates that our tracker enjoys a strong distinction ability. This demonstrates that our approach does not introduce excessive background noise and avoids template contamination. In the (d) sequence, where there exist multiple interfering factors including complex illumination, occlusion, variation in target size, and complex background, our IIMF model achieves continuous tracking of the target. This demonstrates that our model enjoys a high tolerance to various extreme conditions. These results indicates our proposed IIMF tracker enjoys high performance in various challenging scenarios, which makes our tracker a robust and accurate one.

## 5 CONCLUSION

In this paper, we introduced a novel RGBT tracking framework that enhances interactive information handling and minimizes information loss by leveraging a cross-attention mechanism to aggregate modality-shared information and modality-specific information for joint learning and interaction. Utilizing the capabilities of Vision Transformers (ViT), our approach simplifies the complexity of cross-modal interactions by embedding rich contextual information directly into ViT, avoiding the need for intricate fusion techniques between RGB and TIR modalities. Additionally, we propose the MMFE and CAMIA modules, which may inadvertently introduce redundant data and background noise, potentially compromising tracking performance. Future work will focus on refining our information aggregation processes to enhance the efficiency and effectiveness of our tracking framework.

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
