# OpenReview forum: "Simplifying Cross-modal Interaction via Modality-Shared Features for RGBT Tracking"
_acmmm.org/ACMMM/2024/Conference — MM2024 Poster_

### Official Review · Reviewer_rSz5 · 2024-05-23

**Rating:** 2
**Confidence:** 4

**Summary:**

This article introduces two methods of modal interaction and the utilization of complementary information, one is directly concatenate features, and the other is the use of fusion templates as intermediaries to achieve modal interaction. Subsequently, the shortcomings of these two methods are pointed out and a new interaction method is proposed, which uses the MMFE module to extract modality specific and shared information, and the CAMIA module to further extract modality shared information. Finally, the effectiveness of the method was validated on three mainstream RGBT datasets.

**Strengths:**

1. It is pointed out that the previous method (TBSI) of using fusion templates for modal interaction can easily make the model overly focus on modal shared information and weaken modal specified information.

2. The proposal of MMFE module and CAMIA module fully utilizes modality specific information and modality shared information to improve tracking performance.

**Limitations:**

1. Overall, this work is more like an extension of TBSI and is more appropriately submitted to a journal.

2. The title is to simplify cross-modal interactions, but the designed interactions do not seem to be simplified or even more complex. In addition, the experiments are insufficient and do not give specific metrics of the interaction module, such as the number of parameters, the amount of calculation, the final tracking efficiency, and so on.

3. The proposed method shows significant performance improvement compared to previous works on the LasHeR dataset, but the improvement on the other two datasets is not very significant or even slightly decreased. It is possible that the model is overfitting.

4. After the MMFE module, the network structure has become three branches, increasing the number of parameters and computation, which may affect the inference speed.

5. Table 3 compares different methods on the RGBT210 dataset, however, the header is written as RGBT234.

6. SiamAfb in Table 2 has also been evluated on the RGBT210 dataset, but it is not listed in Table 3.

**Suitability:**

2

---

### Official Review · Reviewer_JfUn · 2024-05-24

**Rating:** 4
**Confidence:** 4

**Summary:**

This paper proposed the Integrating Interaction into Modality-shared Features with ViT (IIMF) framework. This framework simplifies cross-modal interaction using shared, RGB-specific, and TIR-specific branches, with a Vision Transformer (ViT) facilitating inter-modal interaction. In addition, Cross-Attention-based Modality-shared Information Aggregation (CAMIA) module further enhances shared information aggregation.

**Strengths:**

## #1. The paper presents a clear and well-motivated problem statement.
## #2. The proposed framework is innovative, with a structured approach to handle cross-modal interactions.
## #3. Experiments on three benchmark datasets demonstrate the method's state-of-the-art performance.

**Limitations:**

### 1. The ablation study presented in the paper is somewhat simplistic. A more detailed analysis of the contributions of different modules would be beneficial. Specifically, there is a need for further justification regarding the effectiveness of the MMFE (Modality-shared, Modality-specific Feature Extraction) module. How do the separated modality-shared, visible-specific, and infrared-specific features contribute to the overall performance? It would be helpful to include feature-level visualizations to demonstrate their effectiveness. Additionally, comparisons with other fusion methods would strengthen the argument for the proposed approach.

### 2.  The results in Table 2 show that the proposed method significantly outperforms SiamAfb in terms of success but lags in precision. This discrepancy warrants further explanation. Is the lower precision indicative of less accurate localization, or does it relate to the ambiguity in ground truth annotations?

### 3.  There are some typographical errors in the manuscript, specifically in the captions of Table 2 and Table 3. Line 790 and the tables indicate results for RGBT210 and RGBT234, but there are inconsistencies in the labeling. Both tables seem to present results for RGBT234, which needs to be corrected for clarity and accuracy.

**Suitability:**

3

---

### Official Review · Reviewer_TRdW · 2024-05-26

**Rating:** 5
**Confidence:** 3

**Summary:**

This paper introduces a modality-shared feature branch by interacting with cross-modal information. This branch and two modality-specific feature branches form a three-branch network. The three-branch network validates the effectiveness of this method on multiple datasets to a certain extent.

**Strengths:**

1.This paper offers a novel MMFE module to disentangle features into modality-shared and modality-specific components.
2.This paper achieves state-of-the-art performance on LasHeR benchmarks.
3.This paper is clear and easy to understand, with a good balance of text and illustrations.

**Limitations:**

1.The connection between related work and the topic of the paper is not tight enough, and the key issues that this paper would like to emphasize are not highlighted.
2.The first contribution, " simplified RGB-T Tracking method," lacks experimental data support.
3.The experimental analysis is insufficient. This method doesn’t achieve state-of-the-art performance in RGBT234 and RGBT210 datasets, which is inconsistent with what is stated in the abstract. The comparison of methods on partial datasets is insufficient and too one-sided.
4.The paper contains a few spelling errors and errors in some tables, such as Table 3.

**Suitability:**

2

---

### Meta-Review · Area_Chair_bpqV · 2024-06-30

**Recommendation:** Accept (Poster)
**Confidence:** 4

**Metareview:**

The initial reviewer ratings for the submission include two weak accept and one borderline reject. Reviewers in general appreciated the technique contribution to the cross-modality tracking. Concerns were raised on motivation, clarity and alginment with ACM MM. The rebuttal was checked by reviewers and they maintained the ratings in the final decision. We considered all reviews and author feedback, and feel that the paper reaches the ACM MM standard and would recommend acceptance.